# Effectiveness of Online Mindfulness-Based Intervention (iMBI) on Inattention, Hyperactivity–Impulsivity, and Executive Functioning in College Emerging Adults with Attention-Deficit/Hyperactivity Disorder: A Study Protocol

**DOI:** 10.3390/ijerph18031257

**Published:** 2021-01-30

**Authors:** Kai-Shuen Pheh, Kit-Aun Tan, Normala Ibrahim, Sherina Mohd Sidik

**Affiliations:** 1Department of Psychology and Counselling, Universiti Tunku Abdul Rahman, Kampar 31900, Malaysia; 2Department of Psychiatry, Faculty of Medicine and Health Sciences, Universiti Putra Malaysia (UPM), Serdang 43400, Malaysia; tanka@upm.edu.my (K.-A.T.); normala_ib@upm.edu.my (N.I.); sherina@upm.edu.my (S.M.S.)

**Keywords:** ADHD, mindfulness, adult

## Abstract

Attention-deficit/hyperactivity disorder (ADHD), a common neurodevelopmental disorder, often persists into adulthood. In Malaysia, the prevalence rate of hyperactivity symptoms is highest among Chinese Malaysians. There are limited evidence-based treatment options targeting the core symptoms of ADHD, as well as executive functioning. In addition, conventional psychotherapeutic approaches for adults with ADHD have been found to be highly labor-intensive. The present study will evaluate the effectiveness of an online mindfulness-based intervention to reduce inattention and hyperactivity–impulsivity and improve executive functioning among Chinese Malaysian college emerging adults with ADHD. Informed by established literature, we will design an 8-week online mindfulness-based intervention (i.e., iMBI). We will conduct a two-arm randomized controlled trial comparing an iMBI plus treatment-as-usual group (*n* = 54) and an enhanced treatment-as-usual control group (*n* = 54). Outcome measures of inattention, hyperactivity–impulsivity, and executive functioning will be collected at baseline, immediately post-intervention, and 1-month post-intervention. The findings of the present study will not only demonstrate the implementation of iMBI as a new treatment modality but also inform practitioners on the effectiveness of iMBI in reducing the burden of adults living with ADHD.

## 1. Introduction

Attention-deficit/hyperactivity disorder (ADHD) is one of the most documented neurodevelopmental disorders in a clinical setting. The *Diagnostic and Statistical Manual of Mental Disorders* (5th ed.; DSM-5) [1] describes ADHD as bidimensional in that it has persistent patterns of inattention and/or hyperactivity–impulsivity leading to significant functional or developmental impairment and executive dysfunctions [2]. ADHD is often regarded as a childhood developmental disorder. However, it may persist into adulthood and have substantial psychiatric comorbidity [3,4,5]. From longitudinal birth cohort studies, it has been estimated that the adult ADHD prevalence rates range from 2% to 6% [6,7,8].

In Malaysia, it was reported that 4.6% of children aged 5 to 15 years old had significant symptoms of hyperactivity [9]. It was also reported that the prevalence rates of the hyperactivity problem were 5.8% among Chinese Malaysian, 4.7% among Malay Malaysian, and 5.3% among Indian Malaysian children [9]. While little epidemiological data are known about adult ADHD in Malaysia, it is reasonable to speculate that the prevalence of such neurodevelopmental disorder among Chinese Malaysian adults would continue to be high, as we inspect the prevalence rate of the hyperactivity problem among Chinese Malaysian children—most individuals’ ADHD symptoms may persist to adulthood.

### 1.1. Emerging Adulthood and ADHD

Emerging adulthood is a distinct developmental phase often taking place from 18 to 25 years old [10]. Issues and challenges that arise during emerging adulthood have received extensive attention from researchers in the past 2 decades. For instance, numerous studies have investigated the demographic status, subjective meaning of adulthood, and identity exploration among emerging adults [10,11,12,13]. Among other issues, psychopathology and mental health problems among emerging adults have been found to be particularly remarkable [14]. Since 2006, late adolescents and emerging adults have had the highest psychiatric morbidity in Malaysia [9,15]. To this end, individuals growing up with ADHD are expected to encounter even more difficulties than their typically developed peers. This notion is supported by past studies indicating that individuals with ADHD had more psychiatric comorbidities [16], greater involvements in risky behaviors [17], higher risk for sexual victimization [18], greater cognitive failure [19], and poorer quality of life [20] as compared with their non-ADHD counterparts.

### 1.2. The Present Study

Based on insights derived from recent empirical research, the present study is devoted to addressing at least five research gaps. These are (1) management of ADHD to improve core symptoms, (2) promotion of non-pharmacological interventions, (3) accessibility of therapy, (4) mindfulness-based intervention as a potential treatment choice, and (5) online intervention as a new treatment modality.

#### 1.2.1. Management of ADHD to Improve Core Symptoms

ADHD is a complex neurodevelopmental disorder contributed to by multiple genetic and environmental risk factors. The neurocognitive deficit theories of ADHD (e.g., executive dysfunction, delay aversion, and dual pathway theories) [21] and neurobiological basis of ADHD [22] have been widely recognized by researchers and clinicians. The theory of impairment of executive functioning proposes a set of neurocognitive strategies leading to the attainment of goals. According to this theory, cognitive deficits such as response inhibition, vigilance, working memory, and planning are most prominent in people with ADHD [23]. However, meta-analytic reviews found that the effect size of existing treatments (e.g., methylphenidate and working memory training) on these executive skills was, at best, small to moderate [24,25]. Such theory suggests a critical need for developing novel and innovative treatments to promote executive functioning. To this end, the present study will focus on inattention, hyperactivity–impulsivity, and executive functioning (in particular, working memory and inhibition) as treatment outcome variables.

#### 1.2.2. Promotion of Non-Pharmacological Interventions

In addition, management of ADHD remains a debatable topic in both scientific and public communities. The National Institute for Health and Care Excellence [26] recommends pharmacological treatments for adults with ADHD who have failed to respond to environmental modifications. Although the efficacy of psychostimulants on ADHD individuals has been well established through numerous high-quality studies [27,28], a number of barriers to pharmacological intervention exist. Individuals with ADHD may have difficulty enjoying the benefits of optimized treatment as a result of nonadherence to ADHD medication [29] and intolerance of adverse effects [30]. In addition, O’Callaghan and Sharma [20] argued that pharmacological therapies may not sufficiently address the psychosocial needs of people with ADHD. Thus, there are ongoing efforts to search for a novel non-pharmacological treatment to maximize desirable clinical outcomes in people with ADHD. The present study will contribute to the limited existing body of knowledge on non-pharmacological interventions by developing and evaluating an online psychotherapeutic program.

#### 1.2.3. Accessibility of Therapy

In Malaysia, non-pharmacological management of ADHD, especially in treatment-resistant cases, is often delivered by clinical psychologists [31]. As reported by the Ministry of Health [32], only 0.03 clinical psychologist (per 1,000,000 population) and 0.01 clinical psychologist (per 1,000,000 population) are available in West and East Malaysia, respectively. The significant scarcity of clinical psychology human resources represents a substantial barrier to service accessibility. Thus, there is an urgent need to improve the accessibility of non-pharmacological therapies for ADHD. Given the advancement of technology, especially in e-mental health, online psychological intervention could be a promising approach to narrow the accessibility gap [33].

#### 1.2.4. Mindfulness-Based Intervention as a Potential Treatment Choice

Mindfulness-based intervention (MBI) is a relatively new psychological intervention modality in modern psychology. The frameworks of this evolving psychotherapy technique date back to about 2500 BC in the Eastern wisdom of spirituality [34]. In the practice of modern medicine, mindfulness’s clinical value was first proven in chronic back pain management in the 1970s [35] and in a wide range of psychiatric disorders in recent years [36,37]. The rationale of employing a mindfulness-based intervention to improve symptoms in emerging adults with ADHD relies on growing evidence of attention-related neuropathway changes resulting from mindfulness practices [38,39,40,41,42,43]. Several empirical studies have indicated that training on mindfulness may also enhance metacognitive processing, where individuals command the skills needed to intentionally self-monitor and regulate their attention and perception [44,45]. Mindfulness training such as mindful breathing could activate neural pathways that are beneficial to attention and executive functions [39]. After a 9-month follow-up, a significant increase in cortical thickness was observed in the right prefrontal cortex extending to the anterior cingulate cortex and in the bilateral occipital regions extending to the inferior temporal cortices among participants receiving mindfulness-based intervention [46]. The said brain regions are known to be associated with attention and executive functioning—the core deficits of people with ADHD. To further support this premise, findings from a recent meta-analysis found that the effectiveness of MBI in alleviating ADHD symptoms, reducing depression, and improving executive functioning was moderate [47]. In this regard, MBI is a good candidate as adjunct therapy for adults with ADHD.

#### 1.2.5. Online Intervention as a New Treatment Modality

The Malaysian Communications and Multimedia Commission reported that 90.1% of Malaysian households have Internet access [48]. Given that a psychological treatment gap does exist in Malaysia, where discrepancies between supply and demand and costs of mental health services are large, online intervention may be a viable alternative. Online interventions have several advantages over conventional face-to-face intervention programs in one important way—they provide self-directive, anonymous, low-cost services and are instantly accessible to communities that otherwise may not have access to mental healthcare services. Evidence from a cross-sectional study also suggests that an Internet-delivered mindfulness program is the first choice of young people [49]. For example, cognitive behavioral therapy (CBT) using an online platform was successfully delivered to people with ADHD [50] and its promising results were reported. Just like online CBT, the effectiveness of an Internet-delivered mindfulness-based intervention to improve mental health was reported in a meta-analytic review of randomized control trials [51].

Taken together, the present study will contribute to the limited existing body of knowledge and current psychiatric management of ADHD by delivering a relatively new treatment modality via a web platform, that is, online mindfulness-based intervention (iMBI). Therefore, the primary objective of the present study is to examine the effectiveness of iMBI in reducing inattention and hyperactivity–impulsivity, and in improving executive functioning among Chinese Malaysian college emerging adults with ADHD.

## 2. Materials and Methods

### 2.1. Study Intervention

A mindfulness-based intervention is typically delivered over 8 weekly structured face-to-face sessions. It aims to cultivate nonjudgmental focused attention on experiences in the present moment [34]. In a group format, an MBI session usually lasts for 2 to 3 h and consists of a short lecture on selected topics, formal and informal in-session practices, and weekly home practices. A search procedure was conducted to identify published research and studies examining the effects of mindfulness on adults with ADHD. Published research and studies were identified by searching eight electronic databases: Scopus, PubMed, ScienceDirect, MEDLINE Complete, PsycArticles, Frontiers, SAGE, and China National Knowledge Infrastructure. Keywords such as attention-deficit/hyperactivity disorder, ADHD, mindfulness, and meditation were included in the search strategies. Four key frameworks concerning the effectiveness of mindfulness-based intervention in reducing inattention and hyperactivity–impulsivity and in improving executive functioning based on insights derived from eight recent empirical research studies were obtained [52,53,54,55,56,57,58,59]. These are (1) psychoeducation on ADHD and mindfulness; (2) mindfulness of physical sensations; (3) mindfulness of cognitive, behavioral, and emotional functioning; and (4) consistent practices of formal and informal mindfulness. We will design an eight-session iMBI in compliance with these four key MBI frameworks.

#### 2.1.1. Online Mindfulness-Based Intervention

Appendix A (see Appendix A) illustrates the proposed outline of the iMBI program. In support of Crane et al. [60], the present program will be developed to meet the specific needs of emerging adults with ADHD. In the present study, iMBI will start with psychoeducation about ADHD and definition of mindfulness (i.e., the first key framework). Participants will engage in a structured discussion session to share about their experiences of living with ADHD. Upon orientation to the program, the participants will be introduced to various topics of mindfulness and ADHD, as well as a wide range of in-session online mindfulness exercises (i.e., the second and third key frameworks). Offline assignments (i.e., the forth key framework) that correspond to a weekly theme will be assigned to the participants at the end of each session. Downloadable audio tracks for offline practices will be provided to the participants. The participants are expected to spend around 15 to 30 min to complete the weekly iMBI session. To examine face and content validities, the proposed program will be reviewed by a group of expert panels, including a family medicine specialist, a psychiatrist, two clinical psychologists, and a developmental psychopathologist. Revision and refinement of the program content will be made based on comments and suggestions from panel discussions. Table 1 displays the mapping of iMBI to outcome variables.

#### 2.1.2. Enhanced Treatment-as-Usual Control Group

Participants from the enhanced treatment-as-usual (eTAU) control group will receive the usual care that is provided to adults with ADHD across clinical settings in Malaysia, along with a 30 min online psychoeducation webinar on adult ADHD. The participants will be allowed to initiate, maintain, or discontinue a treatment as they wish. We play no role in determining the treatment plan of the participants. However, we will emphasize the importance of seeking treatment as needed from their respective clinicians throughout the trial. The participants will be required to report both pharmacological and non-pharmacological treatments that they have received through an online questionnaire during the trial period. Data from the eTAU control group participants receiving either mindfulness-based intervention or CBT from their treating clinicians during the trial period will be excluded from statistical analysis to avoid data contamination. The participants from the eTAU control group will receive an invitation to participate iMBI upon completion of data collection.

### 2.2. Study Design

This is a single-blind randomized controlled trial. Participants who fulfil the inclusion and exclusion criteria will be randomly assigned to an 8-week online mindfulness-based intervention plus treatment-as-usual (TAU) group (i.e., iMBI + TAU) or an eTAU control group using a simple randomization technique. Inattention, hyperactivity–impulsivity, and executive functioning of the participants from both the iMBI treatment and eTAU control groups will be assessed at baseline (week 0), immediate post-intervention (week 8), and 1-month follow-up (week 12).

### 2.3. Randomization

A simple randomization technique will be used to allocate eligible participants to either the iMBI treatment + TAU group or the eTAU control group. The present study will use sequentially numbered, opaque, sealed envelopes for randomization [61]. To ensure that the number of participants is balanced between both groups, even if the recruitment of participants ends earlier than expected, a permuted block of four and six will be used. To blind the participants regarding their conditions, they will be informed that they are going to receive an online healthcare program without providing information regarding their treatment condition.

### 2.4. Study Population

As college emerging adults are reportedly facing multiple challenges in life, including mental health problems [16], risky health behaviors [17], and low educational outcomes [62], these challenges could be particularly pertinent to emerging adults with ADHD. Hence, an effective intervention targeting college emerging adults with ADHD is warranted. The study population will be Chinese Malaysian college emerging adults who have met the DSM-5 diagnostic criteria for ADHD, aged 18 to 29 years in Malaysia. Participants will be recruited from universities or colleges in Malaysia.

### 2.5. Sample Size

Sample size estimation is based on a meta-analytic approach. This estimation includes mindfulness-based intervention studies of individuals with ADHD published from 2008 to 2019 [41,52,53,54,55]. Using a power of 80%, *α* = 0.05, and effect size *f* = ½ *d*, the sample size for the present study is estimated. For inattention, Cohen’s *d* ranges from 0.33 to 3.14. For hyperactivity–impulsivity, Cohen’s *d* ranges from 0.24 to 1.32. For executive functioning, Cohen’s *d* ranges from 1.08 to 2.16. For all treatment outcome variables, it is suggested that a sample size of 45 per condition is needed to detect a power of 0.80 and *α* = 0.05. The present study anticipates a 20% attrition rate, so a sample size of 54 participants per condition is needed, totaling 108 participants.

### 2.6. Inclusion and Exclusion Criteria

The six-item Adult ADHD Self-Report Screening Scale for DSM-5 [63] will be employed to identify prospective participants. With respect to the inclusion criteria, participants must have a diagnosis of ADHD established with the Diagnostic Interview for Anxiety, Mood, and Obsessive–Compulsive and Related Neuropsychiatric Disorders (DIAMOND)–ADHD Diagnostic Module [64], must be aged between 18 and 29 years old, and must be proficient in Mandarin. Participants taking psychotropic prescriptions will be allowed to participate if their prescriptions are stable for at least 6 weeks prior to their participation in the present study.

The exclusion criteria for the present study will include (1) history or current presence of nonalcohol and/or nontobacco substance dependence, psychotic illness, bipolar disorder, personality disorders, conduct disorder, chronic suicidal behavior, and self-injurious behavior as reported in the DIAMOND self-report screening questionnaire; (2) nonverbal intellectual ability of percentile rank 25 or below as described in Raven’s Progressive Matrices–Clinical Edition [65]; and (3) received CBT for ADHD or mindfulness training in the past 2 years.

### 2.7. Treatment Outcome Measures

To measure the symptoms of inattention and hyperactivity–impulsivity, the Adult ADHD Self-Report Scale (ASRS v1.1) Symptom Checklist [66] will be used. The ASRS v1.1 is a scale developed to determine the presence and frequency of current ADHD symptoms in adults. The scale, which consists of 18 items, is rated by participants on a 5-point Likert scale ranging from 0 (*never*) to 4 (*very often*). The ASRS v1.1 yields two subscale scores, namely, inattention and hyperactivity–impulsivity.

The Adult Executive Functioning Inventory (ADEXI) [67] will be used to assess the executive functioning outcome in the present study. The ADEXI is a 14-item self-report scale assessing working memory and inhibition in adulthood with a 5-point Likert scale ranging from 1 (*definitely not true*) to 5 (*definitely true*). The ADEXI has a summed score ranging from 14 to 70. Higher scores indicate greater difficulty with executive functioning.

### 2.8. Sociodemographic Information and Clinical Characteristics

Sociodemographic information (e.g., gender, age, and ethnicity) and clinical characteristics (e.g., subtype of ADHD, comorbidity, date/year of first ADHD diagnosis, use of ADHD medication, and level of adherence to ADHD medication) will be collected for descriptive purposes.

### 2.9. Ethical Considerations

The present study was approved by the Scientific Research Ethical Committee of Universiti Tunku Abdul Rahman (UTAR, Ref: U/SERC/93/2019). The present study was registered with ClinicalTrial.gov (NCT04229251). Information sheets and consent forms will be provided for all participants involved in the trial. We will discuss the trial with prospective participants regarding the information provided in the information sheets. Prospective participants will be able to have an informed discussion with us. Written consent from patients willing to participate in the trial will be obtained.

All study-related information will be stored securely at the study site. All participant information will be stored in double-locked file cabinets in areas with limited access. All questionnaires and data collection, process, and administrative forms will be identified by a coded ID number only to maintain participant confidentiality. All documents that contain names or any personal identifiable information, such as informed consent forms, will be stored separately from study records identified by code number. All local databases will be secured with password-protected access systems. Forms, lists, attendance lists, appointment books, and any other listings that link participant ID numbers to other identifying information will be stored in a separate locked file in an area with limited access. Participants’ study information will not be released outside of the study without the written permission of the participants.

### 2.10. Study Procedures

Recruitment of participants will take place in different ways. First, participants from the participating university will be randomly recruited. Second, participants will be recruited from the counselling center at the university. Third, participants will be recruited via campus-based online social media sites (e.g., Facebook) and students’ online portal of the participating university. Last but not least, recruitment presentations at the participating university will also be made to recruit participants. We will start to enroll participants in March 2021 by invitation. The trial is expected to begin in May 2021 and end in December 2021.

Interested individuals will receive a study information sheet and will be invited to a screening session to determine their eligibility. Written informed consent will be obtained from all participants. Eligibility of participants will be assessed using screening and diagnostic measures 2 weeks before the program starts. Data collection will be conducted at a designated laboratory. Participants will be required to complete all study measures. Outcome measures, including attention, hyperactivity, and executive functioning (working memory and inhibition), will be collected at baseline, immediate post-intervention, and 1-month follow-up. Figure 1 presents the Consolidated Standards of Reporting Trials (CONSORT) diagram of the present study.

### 2.11. Data Analysis

Data will be analyzed by using the IBM Statistical Package for the Social Sciences (SPSS) version 27(IBM, Armonk, NY, USA). Multivariate analysis of covariance (MANCOVA) will be employed to assess mean differences of outcome variables at immediate post-intervention and 1-month follow-up between the two groups (i.e., iMBI + TAU vs. eTAU) simultaneously with baseline scores as covariates. Pairwise comparisons will be conducted to identify specific differences between the groups as deemed necessary. Statistical significance is set at *p* < 0.05. The magnitude of the effect size will be reported using Cohen’s *d* value. When necessary, study data will be analyzed using intention-to-treat analysis (ITT). For ITT, the last observation carried forward method will be used to impute missing values or dropouts. We will perform per-protocol analysis to examine the effectiveness of iMBI when implemented in an optimal manner.

## 3. Discussion

The present study would have both theoretical and practical implications. With respect to theoretical implications, the present study may demonstrate the successful application of the executive dysfunction theory of ADHD in a Chinese Malaysian college sample. With respect to practical implications, the present study will inform practitioners on the effectiveness of iMBI and its potential in managing cognitive behavioral outcomes in emerging adults with ADHD. More work is necessary to increase the external validity of iMBI, with some suggestions offered as follows. First, only college students will be recruited in the present study. Given that dropout rates in high school among individuals with ADHD are high, high school ADHD patients may have greater functional impairment compared with college ADHD patients. Future studies examining the effectiveness of iMBI in reducing functional impairment in high school ADHD are needed. Second, participants from the eTAU control group may experience varied treatment-as-usual care, depending on the clinical wisdom of their treating clinicians. Future studies should focus on how varying treatment-as-usual conditions may result in different treatment effects. Last but not least, the present study will exclude individuals with significant comorbidity of psychiatric disturbances who may experience severe symptoms of ADHD. In future studies, recruiting adult ADHD with a different psychiatric comorbid diagnosis may be of benefit.

## 4. Conclusions

The findings of the present study will not only demonstrate the implementation of iMBI as a new treatment modality but also inform practitioners on the effectiveness of iMBI in reducing the burden of adults living with ADHD.

## Figures and Tables

**Figure 1 ijerph-18-01257-f001:**
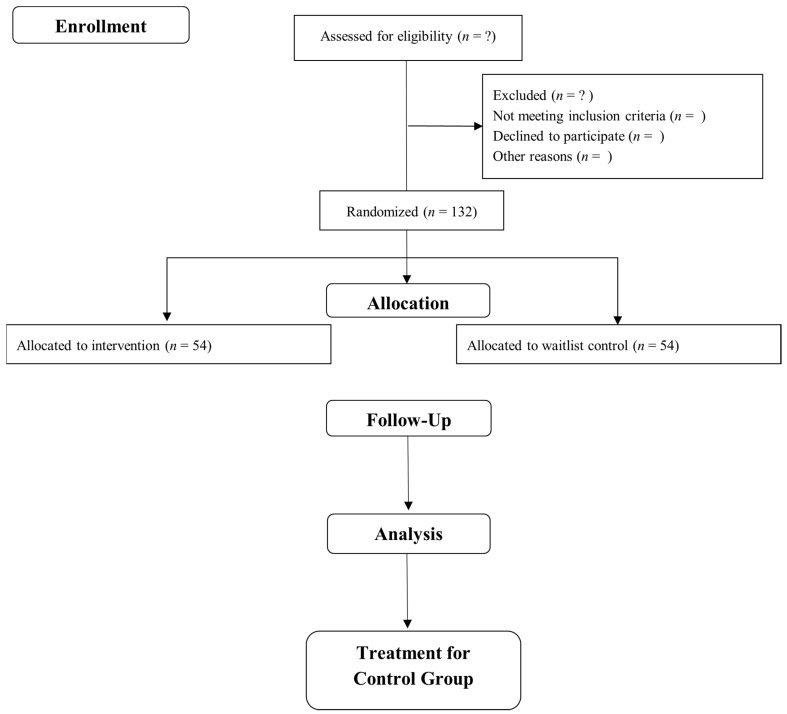
Consolidated Standards of Reporting Trials (CONSORT) flow diagram.

**Table 1 ijerph-18-01257-t001:** Mapping of online mindfulness-based intervention to outcome variables.

Session	Outcome Variables		
	Inattention	Hyperactivity–Impulsivity	Executive Functioning
Session 1: Introduction to ADHD and Mindfulness	✓	✓	✓
Session 2: ADHD through the Lens of Mindfulness	✓	✓	
Session 3: Thinking, Noting, and Feeling	✓	✓	✓
Session 4: Mindful Awareness of Physical Sensation	✓		✓
Session 5: Mindfully Attend to Inattention	✓		✓
Session 6: Time Management with Mindfulness		✓	✓
Session 7: Mindful Awareness of Communication		✓	✓
Session 8: Review and Future Practices			✓
Total Number of Sessions	5	5	7

ADHD is Attention-deficit/hyperactivity disorder.

## Data Availability

Data sharing is not applicable to this study protocol.

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
