# Peer review of "Effectiveness of Online Mindfulness-Based Intervention (iMBI) on Inattention, Hyperactivity–Impulsivity, and Executive Functioning in College Emerging Adults with Attention-Deficit/Hyperactivity Disorder: A Study Protocol"

_ijerph, 2021, doi:10.3390/ijerph18031257_

Round 1

Reviewer 1 Report

It is very interesting study protocol. I look forward for the results.

However, the online method of iMBI / mindfulness training is difficult to provide, and not highly effective in my personal opinion but the only one available in the time of pandemic. So the differences between groups can be very small and not statistically significant. I would increase the number of participants in the groups if it would be possible.

But still - very interesting and well prepared study - congratulations !

I do not reccomend any changes in the present version of the protocol.

Author Response

Response to Reviewer 1 Comments

Comments:

It is very interesting study protocol. I look forward for the results.

However, the online method of iMBI / mindfulness training is difficult to provide, and not highly effective in my personal opinion but the only one available in the time of pandemic. So the differences between groups can be very small and not statistically significant. I would increase the number of participants in the groups if it would be possible.

But still - very interesting and well-prepared study - congratulations!

I do not reccomend any changes in the present version of the protocol.

Responses:

We would like to express our gratitude to your positive response to the manuscript.

We acknowledge that providing online mindfulness-based intervention could be challenging. We concur that a study to investigate the efficacy of iMBI in improving cognitive functioning of individuals with ADHD in the time of pandemic is greatly needed.

If we were to increase the power of the study to .95, there will be an additional need of 66 participants. In view that our study budget has been formally approved by the sponsoring institution, increasing the sample size may not be possible. Should the present study show promising results, we will explore the opportunities of conducting a larger trial in the future.

Reviewer 2 Report

Overall this is a worthy research topic and quite timely at the time of a pandemic when in-person instruction could be risky and have low patient attendance rate. However, a long list of theoretical and methodological issues remain before this can be published as an informative protocol. Below are itemized points:

  1. Line 33, "has persistent patterns of inattention and hyperactivity-impulsivity leading to malfunctioning" This definition could be out of date. Hyperactivity is NOT a defining characteristic, particularly among adult ADHD and hyperactivity does NOT necessarily persist into adulthood. A definition focusing on the developmental issue with executive function will be more proper and update.
  2. Line 42: Given the sociodemographic features of Chinese Malaysians, their "high prevalence rate of ADHD" could be a function of high diagnosis/detection rate, rather than higher rate of genetic risk factors. Thus the inference about Chinese Malaysian adults' ADHD rate could be flawed.
  3. Line 81, it is a bit too vague to say "its’ long-term benefits and adverse effects remain uncertain". There have been solid studies, including systematic review and meta-analyses, about the long-term benefits and adverse effects. The authors need to at least say something more specific about benefits and adverse effects of ADHD drugs. Or, given the context, the authors only need to mention barriers to pharmacological intervention (long-term costs, fear about adverse effect, adherence challenges, etc.).
  4.  Line 94 scholarly literature normally avoids words such as "humongous barrier". Words like "substantial" and "formidable" are more appropriate.
  5. One mechanism MBIs improve ADHD symptoms could be through improving adherence to medication. I strongly suggest that the authors add adherence to ADHD medication either as an outcome measure for the subsample on medication or as an intervening variable that connects the intervention and primary outcomes.
  6. The control group is NOT getting ethically acceptable "usual care". Please give the control group what is standard practice for adult ADHD. I understand the importance of blinding but it is possible to do blinding and a "usual care"/"enhanced usual care". The current plan for the control group is not a care plan for ADHD.
  7. Line 99, the term "Western psychology" is improper. Please use terms like "modern".
  8. Line 169, if "history or current presence of substance dependence" include tobacco and alcohol, the authors might exclude too many otherwise eligible people. One consequence of ADHD is addiction and so the authors might exactly be excluding people this therapy could help.
  9. The analytic plan does not describe models that include covariates/confounding factors. Please describe some mixed-effect models (GEE, GLM, etc.) to present a more sophisticated and comprehensive assessment of the intervention effects.
  10. The Discussion is inadequate. For example, the authors can discuss the limits. One major limit is that ADHD patients have high dropout rates in high schools and low college attendance rates. So by focusing on college populations the authors might have a sample biased toward those least symptomatic.  

Author Response

Thank you for your valuable comments. We take your concerns seriously and have tried our best to address your points in detail. We have made the following changes in response to your comments. Please note that page and line numbers may appear differently on different computers.

Point 1: Line 33, "has persistent patterns of inattention and hyperactivity-impulsivity leading to malfunctioning" This definition could be out of date. Hyperactivity is NOT a defining characteristic, particularly among adult ADHD and hyperactivity does NOT necessarily persist into adulthood. A definition focusing on the developmental issue with executive function will be more proper and update.

Response 1: We have revised the sentence. It now reads:

The Diagnostic and Statistical Manual of Mental Disorders (5th ed.; DSM–5) [1] described ADHD as bi-dimensional in that it has persistent patterns of inattention and/or hyperactivity-impulsivity leading to significant functional or developmental impairment and executive dysfunctions [2].” (see page 1, lines 36–39).

Point 2: Line 42: Given the sociodemographic features of Chinese Malaysians, their "high prevalence rate of ADHD" could be a function of high diagnosis/detection rate, rather than higher rate of genetic risk factors. Thus, the inference about Chinese Malaysian adults' ADHD rate could be flawed.

Response 2: The sentence “The high prevalence rate of ADHD among Chinese Malaysia” was cited from the National Health and Morbidity Survey 2015 which included a nationally representative sample of 5182 Malaysian children. We agree that this sentence is not clear, so we have revised the sentence. It now reads:

“It was also reported that the prevalence rate of hyperactivity problem was 5.8% for Chinese Malaysian, 4.7% for Malay Malaysian, and 5.3% for Indian Malaysian children [9].(see page 2, lines 45–47)

Point 3: Line 81, it is a bit too vague to say "its’ long-term benefits and adverse effects remain uncertain". There have been solid studies, including systematic review and meta-analyses, about the long-term benefits and adverse effects. The authors need to at least say something more specific about benefits and adverse effects of ADHD drugs. Or, given the context, the authors only need to mention barriers to pharmacological intervention (long-term costs, fear about adverse effect, adherence challenges, etc.).

Response 3: We agree that the sentence is not clear. We have revised the sentence, it now reads:

“Although the efficacy of psychostimulants on people ADHD have been well established through numerous high quality studies [26-27], a number of barriers to pharmacological intervention exist. Individuals with ADHD may have difficulty enjoying the benefits of optimized treatment as a result from non-adherence to ADHD medication [29] and intolerance with adverse effects [30]. (see pages 2–3, lines 87–93)

Point 4: Line 94 scholarly literature normally avoids words such as "humongous barrier". Words like "substantial" and "formidable" are more appropriate.

Response 4: Corrections were made as suggested by the reviewer. The sentence now reads:

“The significant scarcity of clinical psychology human resource represents a substantial barrier to service accessibility.” (see page 3, line 104)

Point 5: One mechanism MBIs improve ADHD symptoms could be through improving adherence to medication. I strongly suggest that the authors add adherence to ADHD medication either as an outcome measure for the subsample on medication or as an intervening variable that connects the intervention and primary outcomes.

Responses 5: We acknowledge that treatment adherence is an important variable to be considered in this study. In this regard, we will collect data of treatment adherence, along with other siociodemographic and clinical characteristics for descriptive purposes. We added the following information:

“Sociodemographic information (e.g., gender & age) and clinical characteristics (e.g., subtype of ADHD, comorbidity, the date/year of first ADHD diagnosis, use of ADHD medication, & level of adherence to ADHD medication) will be collected for descriptive purposes.” (see page 7, lines 271 - 274)

We decided not to add adherence to ADHD medication as an outcome measure nor intervening variable as we would like to target cognitive and behavioural outcomes instead of pharmacological outcomes such as ADHD treatment adherence. Given what we know from the literature, targeting pharmacological outcomes would require a different behavioural change intervention approach (Kamimura-Nishimura et al., 2019).

Point 6: The control group is NOT getting ethically acceptable "usual care". Please give the control group what is standard practice for adult ADHD. I understand the importance of blinding but it is possible to do blinding and a "usual care"/"enhanced usual care". The current plan for the control group is not a care plan for ADHD.

Response 6: We have revised the design of control group by taking ADHD standard practice into consideration. It is now known as enhanced treatment-as-usual (eTAU) control group. We have also added additional information concerning the implementation of eTAU control group:

“Participants from eTAU control group will receive the usual care that provided adults with ADHD across clinical settings in Malaysia, along with a 30-minute online psychoeducation webinar on adult ADHD. Participants will be allowed to initiate, maintain or discontinue a treatment as they wish. We play no role in determining treatment plan of the participants. However, we will emphasise the importance of seeking treatment as needed from their respective clinicians throughout the trial. Participants will be required to report both pharmacological and/or non-pharmacological treatments that they received through an online questionnaire during the trial period. Data of eTAU control group participants receiving either mindfulness-based intervention or cognitive behaviour therapy from their treating clinicians during the trial period will be excluded from statistical analysis, to avoid data contamination. Participants from the eTAU control group will receive invitation to participate iMBI upon completion of data collection.” (see page 5, lines 197–207)

Point 7: Line 99, the term "Western psychology" is improper. Please use terms like "modern".

Response 7: Corrections were made as suggested by the reviewer. The sentence now reads:

“Mindfulness-based Intervention (MBI) is a relatively new psychological intervention modality in Western modern psychology.” (see page 3, line 111)

Point 8: Line 169, if "history or current presence of substance dependence" include tobacco and alcohol, the authors might exclude too many otherwise eligible people. One consequence of ADHD is addiction and so the authors might exactly be excluding people this therapy could help.

Response 8: We will recruit individuals with tobacco and alcohol use and will exclude prospective participants with the history of other forms of illicit drugs dependence. We have clarified the exclusion criteria, it now reads:

“Exclusion criteria for the present study will include: (1) history or current presence of non-alcohol and/or non-tobacco substance dependence, psychotic illness, bipolar disorder, personality disorders, conduct disorder, chronic suicidal, or self-injurious behaviour as reported in the DIAMOND self-report screening questionnaire, (2) non-verbal intellectual ability of percentile ranks 25 or below as described in the Raven’s Progressive Matrices–Clinical Edition, and (3) received cognitive-behavioural therapy (CBT) for ADHD or mindfulness training in the past 2 years.” (see page 7, lines 251–257)

Point 9: The analytic plan does not describe models that include covariates/confounding factors. Please describe some mixed-effect models (GEE, GLM, etc.) to present a more sophisticated and comprehensive assessment of the intervention effects.

Response 9: We agree that a more sophisticated and comprehensive assessment of the intervention effects is needed. In this regard, we would like to perform multivariate analysis of covariance (MANCOVA) by considering baseline outcome measurements as the covariates.

We did not consider other multivariate analyses such as generalized estimating equations (GEE) for one main reason. As the present study is not a multi-centre RCT by design. Ukoumunne, Carlin, and Guilliford (2007) argue that GEE may underestimate standard error of the treatment effect when the number of centres is relatively small as in the present study.

The Data Analytic Plan section now reads:

The data will be analysed by using the Statistical Package for the Social Sciences (SPSS) version 27. The multivariate analysis of covariance (MANCOVA) will be employed to assess mean differences of outcome variables at immediately post-intervention, and 1-month follow-up between the two groups (i.e. iMBI + TAU vs. eTAU) simultaneously with baseline scores as covariates. Pairwise comparisons will be conducted to identify specific differences between the groups as it deems necessary.” (see page 9, line 350–359)

Point 10: The Discussion is inadequate. For example, the authors can discuss the limits. One major limit is that ADHD patients have high dropout rates in high schools and low college attendance rates. So by focusing on college populations the authors might have a sample biased toward those least symptomatic.

Response 10: We have revised the Discussion by highlighting the issue of generalizability as study limitations. It reads: 

“Nonetheless, a few study limitations surrounding the issue of generalizability should be noted. First, only college students will be recruited in the presents study. Given that dropout rates in high school among individuals with ADHD were high, high school ADHD patients may have greater functional impairment compared with college ADHD patients. Secondly, participants from eTAU control group may experience varied treatment-as-usual care, depending on the clinical wisdom of their treating clinicians. Third, this study will exclude individuals with significant comorbidity of psychiatric disturbances, who may experience more severe symptoms of ADHD.” (see page 8, lines 366–373)

Thank you again for all the suggestions and comments. We hope our revisions meet your approval.

References

Kamimura-Nishimura, K. I., Brinkman, W. B., & Froehlich, T. E. (2019). Strategies for improving ADHD medication adherence. Current Psychiatry, 18(8), 25.

Ukoumunne, O. C., Carlin, J. B., & Gulliford, M. C. (2007). A simulation study of odds ratio estimation for binary outcomes from cluster randomized trials. Statistics in Medicine, 26(18), 3415–3428.

Reviewer 3 Report

Dear authors, dear editors,

thanks for asking me to review this protocol. The research idea sounds interesting: the authors propose a protocol for a study investigating the effects of a MBI on symptoms of ADHD, in adults.

After carefully reviewing the paper, I am suggesting the authors to apply minor revisions and re-submit it, before publication in this journal (I am looking forward to follow the outcomes of this study).

The study proposal is quite detailed, especially in regards to the methodology and the analytic plan. I advise the authors to carefully check English language and spelling: in some sections, in fact, there are minor grammatical and spelling errors.

Detailed comments

-Please, add country to affiliation 1

-Abstract: please, clarify the sentence "In Malaysia, the prevalence rate of hyperactivity symptoms was highest in Chinese", do you mean that the prevalence rate of ADHD amongst Chinese citizens living in Malaysia was higher than Malaysian citizens?

-L40. "Noteworthy, the prevalence rate of hyperactivity problem was highest among Chinese children". Please, report the prevalence rate among Chinese children. Moreover, why are you referring to children when the main focus of the paper is on adults? Are there any differences in prevalence of ADHD among Chinese and Malaysian adults? This should be reproted in the paper, if authors think that it is relevant.

-Although the authors are presenting some references to support the use of mindfulness-based therapy in ADHD (L 113), the rationale for proposing such intervention is not fully clear to me. More specifically, could the authors discuss a bit more why they think MBI is likely to improve executive functions in adults with ADHD (paragraph 1.2.1 and L 131)? Moreover, could the authors specify why they have not planned to compare the effects of such intervention and the effects of gold-standard pharmacological intervention? I think this should be expressed more clearly to the reader.

-Moreover, are the authors trying to collect additional objective measures, for example in relation to physiological/autonomic arousal? There is some recent literature proposing that reduced physiological arousal and functioning of the autonomic nervous system might be core in ADHD symptomatology (see https://doi.org/10.1016/j.neubiorev.2019.11.001). Considering the literature showing that mindfulness is likely to have some effects on stress and physiological mechanisms, do the authors think that this would be worth to be investigated in their sample?

-L81. Please correct its’ in its, and correct "may not be sufficiently addressed" in "may not sufficiently address".

-Materials and methods: I think the MBI should be presented at first, in this paragraph, before presenting the study design. For example it is not fully clear what " immediate post-intervention" means, since the length of the intervention has not been specified yet.

-Study population: Could the authors specify why they would like to recruit from universities or colleges? This is not fully clear to me.

-L204. Since this intervention has not yet happened, please use the future tense, and not the past.

-L222. Will the Control Group Programme include any information about ADHD, from a psycho-educational perspective?

-Study Procedures. Could the authors indicate when they plan to start recruitment and when they think the study will be completed?

I am looking forward to review a second draft.

Many regards,

A

Author Response

Thank you for your thoughtful review of our manuscript. We take your concerns seriously and have tried our best to address your points in detail. Changes made in response to your comments and suggestions have resulted in a stronger manuscript. We hope our revision meets your approval. Please note that page and line numbers may appear differently on different computers.

Point 1: Please, add country to affiliation 1

Response 1: Corrections were made as suggested by the reviewer.

Point 2: Abstract: please, clarify the sentence "In Malaysia, the prevalence rate of hyperactivity symptoms was highest in Chinese", do you mean that the prevalence rate of ADHD amongst Chinese citizens living in Malaysia was higher than Malaysian citizens?

Response 2: The term “Chinese” refers to Chinese Malaysians. They are Malaysian citizens who are descendants of ethnic Chinese. To improve clarity, we have rephrased the sentence. It reads:

In Malaysia, the prevalence rate of hyperactivity symptoms was highest in Chinese Malaysian.” (see page 1, line 17)

Point 3: L40. "Noteworthy, the prevalence rate of hyperactivity problem was highest among Chinese children". Please, report the prevalence rate among Chinese children. Moreover, why are you referring to children when the main focus of the paper is on adults? Are there any differences in prevalence of ADHD among Chinese and Malaysian adults? This should be reported in the paper, if authors think that it is relevant.

Response 3: As suggested by the reviewer, we have revised the sentence to render greater clarity. It now reads:

In Malaysia, it was reported that 4.6% of children aged from 5 to 15 years old had significant symptoms of hyperactivity [9]. It was also reported that the prevalence rate of hyperactivity problem was 5.8% for Chinese Malaysian, 4.7% for Malay Malaysian, and 5.3% for Indian Malaysian children [9].” (see page 2, lines 44–47)

Point 4: Although the authors are presenting some references to support the use of mindfulness-based therapy in ADHD (L 113), the rationale for proposing such intervention is not fully clear to me. More specifically, could the authors discuss a bit more why they think MBI is likely to improve executive functions in adults with ADHD (paragraph 1.2.1 and L 131)? Moreover, could the authors specify why they have not planned to compare the effects of such intervention and the effects of gold-standard pharmacological intervention? I think this should be expressed more clearly to the reader.

Response 4: We have now added information on the potentials of mindfulness-based intervention in altering neural pathways thus improving executive functions in adults with ADHD. It reads:

“Mindfulness training such as mindful breathing could activate neural pathways that are beneficial to attention and executive functions. After a 9-month follow-up, a significant increase in cortical thickness was observed in the right prefrontal cortex extending to anterior cingulate cortex and in bilateral occipital regions extending to inferior temporal cortices among participants receiving mindfulness-based intervention. The said brain regions were known to be associated with attention and executive functioning—the core deficits of people with ADHD. To further support this premise, findings from a recent meta-analysis have found that the effect of MBI in alleviating ADHD symptoms, reducing depression, and improving executive functioning was moderate. To this end, MBI is a good candidate of adjunct therapy for adults with ADHD.” (see page 3, lines 119–128)

We have also revised the design of control group by taking gold-standard pharmacological intervention into consideration. We added this information in the main text, it reads:

“Participants from eTAU control group will receive the usual care that provided adults with ADHD across clinical settings in Malaysia, along with a 30-minute online psychoeducation webinar on adult ADHD. Participants will be allowed to initiate, maintain or discontinue a treatment as they wish. We play no role in determining treatment plan of the participants. However, we will emphasise the importance of seeking treatment as needed from their respective clinicians throughout the trial. Participants will be required to report both pharmacological and/or non-pharmacological treatments that they received through an online questionnaire during the trial period. Data of eTAU control group participants receiving either mindfulness-based intervention or cognitive behaviour therapy from their treating clinicians during the trial period will be excluded from statistical analysis, to avoid data contamination. Participants from the eTAU control group will receive invitation to participate iMBI upon completion of data collection.” (see page 5, lines 197–207)

Point 5: Moreover, are the authors trying to collect additional objective measures, for example in relation to physiological/autonomic arousal? There is some recent literature proposing that reduced physiological arousal and functioning of the autonomic nervous system might be core in ADHD symptomatology (see https://doi.org/10.1016/j.neubiorev.2019.11.001). Considering the literature showing that mindfulness is likely to have some effects on stress and physiological mechanisms, do the authors think that this would be worth to be investigated in their sample?

Response 5: We agree that collecting additional objective measures could add value to the present study. However, there have been mixed reviews surrounding the ecological validity of ADHD objective measures (see Barkley, 1991; Barkley & Fischer, 2011). In a systematic review and meta-analysis, it was reported that only small effect size of mean differences was found in task-related heart rate variability between individuals with ADHD and healthy controls (see Robe, Dobrean, Cristea, Păsărelu, & Predescu, 2019).

We have also considered collecting neuropsychological performance data via the Test of Variables of Attention (T.O.V.A.). However, the T.O.V.A. is laboratory-based and requires in-person attendance. The current physical distancing policies that being implemented in Malaysia may pose difficulty administering such test.

Point 6: L81. Please correct its’ in its, and correct "may not be sufficiently addressed" in "may not sufficiently address".

Response 6: We have corrected the sentence as suggested by the reviewer. It now reads:

In addition, O’Callaghan and Sharma [20] argued that pharmacological therapies may not sufficiently address psychosocial needs of people with ADHD.” (see page 3, lines 93-95)

Point 7: Materials and methods: I think the MBI should be presented at first, in this paragraph, before presenting the study design. For example it is not fully clear what " immediate post-intervention" means, since the length of the intervention has not been specified yet.

Response 7: We have reorganized the Material and Methods section as suggested by the reviewer. We have also defined the term immediate post-intervention by mentioning its time point.  It reads:

This is a single blind randomised controlled trial. Participants who fulfilled the inclusion and exclusion criteria will be randomly assigned to an eight-week online mindfulness-based intervention group plus TAU (i.e., iMBI + TAU) or eTAU control group using simple randomisation technique. Participants’ inattention, hyperactivity-impulsivity, and executive functioning from both iMBI treatment and eTAU control groups will be assessed at baseline (Week 0), immediate post-intervention (Week 8), and 1-month follow-up (Week 12).” (see page 6, line 209-215)

Point 8: Study population: Could the authors specify why they would like to recruit from universities or colleges? This is not fully clear to me.

Response 8: We have added a paragraph to justify our study population. It reads:

“As college emerging adults are reportedly facing multiple challenges in life, including mental health problems[16], risky health behaviours [17], and low educational outcomes [54], these challenges could be particularly pertinent to emerging adults with ADHD. Hence, an effective intervention targeting college emerging adults with ADHA is warranted. The study population will be Chinese Malaysian college emerging adults, who met DSM-5 diagnostic criteria of ADHD, aged from 18 to 29 years in Malaysia.” (see page 6, lines 226–233)

Point 9: L204. Since this intervention has not yet happened, please use the future tense, and not the past.

Response 9: We have revised the sentence, it now reads:

“We will design an eight-session iMBI in compliance with these four key MBI frameworks.” (see page 4, line 164)

Point 10: L222. Will the Control Group Programme include any information about ADHD, from a psycho-educational perspective?

Response 10: We have revised the design of control group. It is now known as enhanced treatment-as-usual (eTAU) control group. We have also added additional information concerning the implementation of eTAU control group:

“Participants from eTAU control group will receive the usual care that provided adults with ADHD across clinical settings in Malaysia, along with a 30-minute online psychoeducation webinar on adult ADHD. Participants will be allowed to initiate, maintain or discontinue a treatment as they wish. We play no role in determining treatment plan of the participants. However, we will emphasise the importance of seeking treatment as needed from their respective clinicians throughout the trial. Participants will be required to report both pharmacological and/or non-pharmacological treatments that they received through an online questionnaire during the trial period. Data of eTAU control group participants receiving either mindfulness-based intervention or cognitive behaviour therapy from their treating clinicians during the trial period will be excluded from statistical analysis, to avoid data contamination. Participants from the eTAU control group will receive invitation to participate iMBI upon completion of data collection.” (see page 5, lines 197–207)

Point 11: Study Procedures. Could the authors indicate when they plan to start recruitment and when they think the study will be completed?

Response 11: As suggested, we have added more detailed information about study procedures:

We will start to enrol participants in March 2021 by invitation. The trial is expected to begin in May 2021 and to end in December 2021.” (see page 9, line 339–340).

Thank you again for all your suggestions and comments. We appreciate the opportunity to revise our work. We hope our revisions meet your approval.

References

Barkley, R. A. (1991). The ecological validity of laboratory and analogue assessment methods of ADHD symptoms. Journal of Abnormal Child Psychology19(2), 149–178.

Barkley, R. A., & Fischer, M. (2011). Predicting impairment in major life activities and occupational functioning in hyperactive children as adults: Self-reported executive function (EF) deficits versus EF tests. Developmental Neuropsychology36(2), 137–161.

Robe, A., Dobrean, A., Cristea, I. A., Păsărelu, C. R., & Predescu, E. (2019). Attention-deficit/hyperactivity disorder and task-related heart rate variability: A systematic review and meta-analysis. Neuroscience and Biobehavioral Reviews99, 11–22.

Round 2

Reviewer 2 Report

Comments have been addressed. Now it is acceptable after proofreading and stylistic check.

Author Response

We would like to express our gratitude to your positive response to the revised manuscript. We have now revised the manuscript for greater clarity and readability. We sincerely hope you will find this revision acceptable.

Reviewer 3 Report

Dear authors, thanks very much for asking me to give a second opinion on your manuscript.

Overall, I think the protocol has now been improved by following the reviewers' suggestions and it is almost ready for being accepted for publication in the journal. The minor points authors should address, are:
  • The manuscript does not flow really well in terms of readability on some sections, and there are several grammatical errors or incorrect sentences in the document (just as examples, "L37 ADHD was primarily regarded as childhood developmental disorders" or L39 "ADHD has comorbid conditions with other psychiatric disorders"). If the authors have some colleagues who are native English-speakers, I suggest asking them to proofread the manuscript; otherwise, they might want to use a proofreading service to improve readability.
  • Besides reporting the Limitations in the discussion, I would suggest the authors to indicate why the specific decisions presented in the protocol (with limitations) have been taken by the research team (a natural question would in fact be: "If you knew about these limitations before conducting the study, why did you design in this way and you did not try to solve them before recruiting your participants?)
Happy to read another draft.

Author Response

We would like to express our gratitude to your positive response to the revised manuscript. We are grateful to your valuable comments. We take your concerns seriously and have tried our best to address your points in detail. Please note that page and line numbers may appear differently on different computers.

Point 1:

The manuscript does not flow really well in terms of readability on some sections, and there are several grammatical errors or incorrect sentences in the document (just as examples, "L37 ADHD was primarily regarded as childhood developmental disorders" or L39 "ADHD has comorbid conditions with other psychiatric disorders"). If the authors have some colleagues who are native English-speakers, I suggest asking them to proofread the manuscript; otherwise, they might want to use a proofreading service to improve readability.

Response 1:

We have revised the entire manuscript for greater clarity and readability, whenever necessary. The following changes have been made:

ADHD is often regarded as a childhood developmental disorder.” (see page 1, line 40)

“However, it may persist into adulthood and have substantial psychiatric comorbidity.” (see page 1, line 40)

Response 2:

Besides reporting the Limitations in the discussion, I would suggest the authors to indicate why the specific decisions presented in the protocol (with limitations) have been taken by the research team (a natural question would in fact be: "If you knew about these limitations before conducting the study, why did you design in this way and you did not try to solve them before recruiting your participants?)

Point 2:

It is difficult if not impossible to address all foreseeable study limitations in light of our research objectives and study design. Instead of discussing our study limitations, we have now proposed a few research directions for future studies:

“More work to increase the external validity of iMBI is necessary, with some suggestions offered as follows. First, only college students will be recruited in the presents study. Given that dropout rates in high school among individuals with ADHD were high, high school ADHD patients may have greater functional impairment compared with college ADHD patients. Future studies examining the effectiveness of iMBI in reducing functional impairment in high school ADHD are needed. Secondly, participants from eTAU control group may experience varied treatment-as-usual care, depending on the clinical wisdom of their treating clinicians. Future studies should focus on how varying TAU conditions may result in different treatment effects. Third, this study will exclude individuals with significant comorbidity of psychiatric disturbances, who may experience severe symptoms of ADHD. In future studies, recruiting adult patients with ADHD with a different psychiatric comorbid diagnosis may be of benefit.” (page 10, lines 383–393)

Once again, we thank you for insightful comments and suggestions. We sincerely hope you will find this revision acceptable.
